# Role of Vanin-1 Gene Methylation in Fat Synthesis in Goose Liver: Effects of Betaine and 5-Azacytidine Treatments

**DOI:** 10.3390/ani15050719

**Published:** 2025-03-03

**Authors:** Xinfang Wang, Yu Shao, Zhi Yang, Haiming Yang, Zhiyue Wang

**Affiliations:** College of Animal Science and Technology, Yangzhou University, Yangzhou 225009, China; mx120220883@stu.yzu.edu.cn (X.W.); shaoyu991201@126.com (Y.S.); hmyang@yzu.edu.cn (H.Y.); dkwzy@263.net (Z.W.)

**Keywords:** *VNN1* gene, DNA methylation, betaine, 5-AZA, liver fat synthesis

## Abstract

Vanin-1 (VNN1) is a pantotheinase that hydrolyzes pantothein to form pantothenic acid (vitamin B5) and cysteamine. Betaine, an essential methyl donor in the body, plays a crucial role in maintaining normal lipid metabolism in animals. Our previous research identified VNN1 as a key candidate gene down-regulated by betaine during lipid synthesis in goose liver. However, the underlying mechanisms of VNN1 in betaine-regulated lipid synthesis in goose liver remain unclear. Therefore, this study aimed to elucidate the role of VNN1 methylation in betaine-regulated lipid synthesis in goose liver by investigating the effects of betaine and 5-Azacytidine (AZA) on serum biochemistry, enzyme activity, VNN1 gene expression, and VNN1 promoter methylation in goslings aged 35–49 days. The results demonstrated that VNN1 methylation plays a significant role in the regulation of lipid synthesis in goose liver by betaine.

## 1. Introduction

Excessive abdominal fat deposition in broiler poultry including goose has become an increasingly prominent issue in the industry [1,2]. Avian lipid metabolism primarily occurs in the liver and is influenced predominantly by lipid metabolism-related genes and transcription factors. Vanin-1 (VNN1) is one of those genes, which expresses a pantetheinase that hydrolyzes pantetheine to produce pantothenic acid (vitamin B5) and cysteamine [3]. *VNN1* plays a crucial role in coenzyme A (CoA) synthesis by regulating pantothenic acid synthesis [4]. The expression and regulation of VNN1 have received extensive attention in fat metabolism. Studies in mice have highlighted the significant role of *VNN1* in liver lipid metabolism [5,6]. For example, Motomura et al. [7] reported that *VNN1* expression was highly elevated in the liver and was significantly increased in the steatotic livers of mice. Moreover, studies have shown that *VNN1* can inhibit cholesterol and fatty acid synthesis, with its expression positively correlated with low-density lipoprotein (LDL) levels [8]. The regulation of *VNN1* expression is closely associated with a liver lipid metabolism-related gene of peroxisome proliferator-activated receptor α (PPARα) [5,9]. The expression of *VNN1* in wild-type mice fasted and treated with lipid-lowering drugs was significantly increased, whereas such increase was not observed in PPARα-deficient mice. PPARα promotes the expression of the *VNN1* gene by binding to a specific binding site on the *VNN1* promoter [10]. Similar regulatory patterns were observed in chickens as well [11].

In addition, DNA methylation as an important epigenetic modification plays an important role in the regulation of fat metabolism [12,13]. Our previous studies have demonstrated that betaine, as a methyl donor, can inhibit the expression of *VNN1* by promoting its DNA methylation, and thereby reduce hepatic fat synthesis and decrease abdominal fat rate in geese [14]. However, the role of *VNN1* gene methylation in goose liver fat synthesis is not clear.

As a pyrimidine nucleoside analogue and a methylation inhibitor, 5-Azacytidine (5-AZA) has a wide range of biological effects [15]. It can be incorporated into DNA or RNA and then connect with DNA methyltransferase (DNMT) via a covalent bond, resulting in DNMT depletion and DNA demethylation [16]. By reducing the level of DNA methylation, 5-AZA regulates gene expression and biological epigenetics in growth and development [17].

Therefore, based on the relationship between betaine and DNA methylation in regulating goose lipid metabolism, we hypothesized that DNA methylation affects the expression of *VNN1*, thereby regulating goose liver lipid synthesis. The DNA methylation status of *VNN1* gene was analyzed by betaine and AZA and their interaction to reveal its regulatory mechanism in fat metabolism. This study will provide new insights into the epigenetic regulation mechanism of goose liver fat metabolism and provide a theoretical basis for poultry nutrition regulation.

## 2. Materials and Methods

### 2.1. Ethics Approval and Consent to Participate

The procedures were performed according to the Regulations for the Administration of Affairs Concerning Experimental Animals of the People’s Republic of China and approved by the Yangzhou University Animal Care and Use Committee (SYXK [Su] IACUC 2021-0027).

### 2.2. Experimental Design and Feeding Management

A total of twenty-eight 35-day-old male Jiangnan white geese with similar body weight (BW) and good health conditions were selected and randomized into four experimental treatments, with seven birds per treatment. All birds were provided by Changzhou Four Seasons Poultry Industry Co., Ltd. (Changzhou, China). The control group was fed a basal diet and injected with normal saline intraperitoneally. The control group was fed a basal diet and intraperitoneally (I.P.) injected with normal saline. The AZA group was fed the same basal diet and treated I.P. with AZA (2 mg/kg). The betaine group was fed the same basal diet but supplemented with betaine (1.2 g/kg) and injected I.P. with normal saline. The AZA+betaine group was fed the same basal diet plus betaine (1.2 g/kg) and treated I.P. with AZA (2 mg/kg). Anhydrous betaine with a purity of 98% was obtained from Beijing Xin Dayang Co., Ltd. (Beijing, China). The 5-AZA with a purity of 98% was purchased from Shanghai Yuanye Biotechnology Co., Ltd. (Shanghai, China). The trial lasted 14 days. The basal diet was a corn–soybean meal formulated primarily as per the *Nutrient Requirements of Poultry* (1994 edition) prepared by the U.S. National Research Council (NRC) [18] and the prior research results of our laboratory [19,20]. The nutritional composition and levels of the experimental diet are shown in Table 1. The geese were housed in a single pen with plastic flooring, characterized by a 2 cm^2^ opening above the ground by 70 cm. Water was available from a semi-open cylindrical plastic tank, while the diet was provided from feeders located on each side of the fence. The geese were kept indoors only for 14 days under the same conditions (temperature: 26.0 °C ± 3.0 °C; relative humidity (RH): 65.5% ± 5.0%; lighting: natural light; housing density: 0.5 m^2^/goose).

### 2.3. Sample Collection

BW was recorded by electronic platform scale (acs-30 Shanghai Qianxin Co., Ltd., Shanghai, China) with the accuracy of 10 g to 30,000 g at 42 and 49 d of age. On the 14th day of the experiment (49 days of age), the geese were fasted for 6 h. Blood samples were then collected from the birds (*n* = 7 per treatment) via a vein of both wings; after a 2000× *g* centrifugation process for 10 min, supernatant serum was separated for biochemistry analysis. After blood sampling, each goose was euthanatized via quick exsanguination as per the animal ethics, and the tissues of liver, kidney, heart, lung, and spleen were collected, then frozen in liquid nitrogen and stored in a refrigerator at −80 °C before gene expression analysis.

### 2.4. Liver Slices

The liver samples stored at −80 °C were taken out, and the liver tissues were cut into pieces of 1 cm × 1 cm × 1 cm, and liver fat slices were prepared on CryoStar NX50 (Thermo Fisher Scientific Inc., Waltham, MA, USA) at −24 °C. The thickness of each slice was controlled to 5 μm, and staining was performed with hematoxylin and eosin. An upright optical microscope (Nikon, Tokyo, Japan) was used for observation and photographs with three 40× fields of view selected. The area and percentage of lipid droplets stained into red in the fields of view were measured by the software Image-Pro Plus 6.0.

### 2.5. Serum Biochemical Indexes

The levels of total cholesterol (TC), triglyceride (TG), low-density lipoprotein (LDL), and high-density lipoprotein cholesterol (HDL-C) in serum were determined with the kits supplied by Nanjing Jiancheng Bioengineering Institute (Nanjing, China).

### 2.6. DNA Methylation-Related Enzyme Activity

The activity of VNN1 enzyme and DNA methyltransferase, as well as the contents of S-adenosyl methionine (SAM) and S-adenosyl homocysteine (SAH) in the liver, were detected by ELISA with the kits purchased from Shanghai Yuduo Biotechnology Co., Ltd. (Shanghai, China).

### 2.7. Bisulfite Sequencing PCR (BSP)

DNA methylation levels in the promoter regions of *VNN1* were quantified using bisulfite sequencing PCR (BSP). The predicted CpG island was found in the gene promoter region (about 2000 bp upstream of the transcription initiation site), and the bisulfite sequencing PCR (BSP) primers were designed using the online Meth Prime software (https://www.methprimer.com/). The primers were (forward primer) TTTATAGTTTTGAGTTAGGGGTGG and (reverse primer) AACCACAAACATTAAACAATCCTAC. DNA was extracted from 4 liver samples using the Mag-MK cfDNA Extraction Kit (Sangon Biotech, Shanghai, China). Upon the verification of the quality and concentration of the DNA samples, polymerase chain reaction (PCR) amplification was carried out after treatment with sodium bisulfite. The unmethylated C in the DNA fragment was transformed to U. The purified PCR products were then ligated to the pUC18-T vector for cloning, under a reaction condition at 18 °C overnight. After the transformation of the recombinant vector, competent bacteria were plated on ampicillin dishes pre-coated with 20 μL 100 mM IPTG and 100 μL 20 mg/mL X-gal, and cultured upside down overnight. Blue-white screening: white colonies growing on IPTG/X-gal dishes were transferred with toothpicks to a broth containing ampicillin and then cultured at 37 °C overnight. The target fragment was TA cloned using pMDTM 19-T Vector Cloning Kit (Takara Bio, Beijing, China), followed by 1% agarose gels (Sangon Biotech, Shanghai, China) electrophoresis for detection and colony PCR for identification. Subsequently, plasmid extraction and sequencing were carried out with M13+/− primers. The software DNAStar 7.1 was used to analyze the gene sequence, software SeqMan Pro 15.0 for sequence alignment, and QUMA for methylation mapping. A total of 12 CpG sites in the *VNN1* gene fragment were detected, including the positions in sequence as follows: 7, 41, 70, 89, 132, 173, 187, 200, 208, 233, 245, 257.

### 2.8. RNA Extraction and Real-Time Polymerase Chain Reaction

Total RNA was extracted with a FastPure Complex Cell/Tissue Total RNA Isolation Kit from Vazyme (Nanjing, China). The RNA was then reversely transcribed into cDNA with HiScript III RT SuperMix for qPCR (+gDNA wiper) from Vazyme (Nanjing, China). The mRNA expression of *VNN1*, fatty acid synthase (FAS), acetyl-CoA carboxylase (ACC), stearoyl-CoA dehydrogenase (SCD), and sterol regulatory element binding protein (SREBP) was detected by ChamQ SYBR qPCR Master Mix kit from Vazyme (Nanjing, China). β-actin was used as an internal reference. The primer sequence is shown in Table 2. The PCR protocol was as follows: initial denaturation for 30 s at 95 °C, followed by 40 cycles each comprising 10 s at 95 °C and 30 s at 60 °C. Four random samples were selected for each group, with each sample analyzed in triplicate. The average cycle threshold (Ct) values were then used for quantification using the 2^−ΔΔCT^ method.

### 2.9. Data Analysis

Microsoft Excel 2023 was used to establish the database, and SPSS 26.0 (IBM, Armonk, NY, USA) was used to analyze the data. Serum and enzyme activity were analyzed by 2 × 2 two-way ANOVA (the injection of AZA and the feeding of betaine and their interaction), and others were analyzed by one-way ANOVA. The differences between treatments were separated by Tukey’s honest significant difference (HSD) test. The statistical significance was established at *p* < 0.05. The results were expressed as the mean and the standard error of the means (SEM).

## 3. Results

### 3.1. Effects of Different Treatments on Body Weight of Geese

As shown in Figure 1, different treatments resulted in no significant effect on the body weight of the geese in the first week (*p* > 0.05). Compared with the control group, the AZA group showed a significant reduction in body weight in the second week (*p* < 0.05). Similarly, a significant difference was observed between the AZA group and the betaine group in the second week (*p* < 0.05). However, no significant difference was seen between the AZA group and the AZA+betaine group (*p* > 0.05).

### 3.2. Effects of Different Treatments on Serum TG, TC, HDL, and LDL of Geese

The results of different treatments on serum TG, TC, HDL, and LDL of the geese are shown in Table 3. Compared with the control group, the AZA group experienced significant effects of the treatment on the serum levels of TG, TC, HDL, and LDL (*p* < 0.05). The level of TC in the betaine group was significantly lower than that in the control group (*p* < 0.05). The level of HDL in the AZA+betaine group was significantly lower than that in the control group (*p* < 0.05).

### 3.3. Effects of Different Treatments on DNA Methylation-Related Enzyme Activities in Liver of Geese

As seen from Table 4, the treatment with AZA demonstrated a significant effect on VNN1 activity, SAM/SAH ratio, and DNMT activity (*p* < 0.05). The treatment with betaine reduced the activity of VNN1 and significantly increased the ratio of SAM/SAH (*p* < 0.05). Compared with the AZA group, the AZA+betaine group increased both SAM/SAH ratio and DNMT activity (*p* < 0.05).

### 3.4. Effects of Different Treatments on Fat Morphology of Goose Liver

The effects of different treatments on the fat morphology of goose liver are shown in Figure 2. Each treatment had no significant effect on liver fat morphology (*p* > 0.05).

However, compared with the control group, more vacuolar degeneration was observed in the liver tissue of the AZA group.

### 3.5. Effects of Different Treatments on DNA Methylation Level (in VNN1 Promoter Region) of Goose Liver

Figure 3 shows that the promoter region of the *VNN1* gene in the AZA group was hypomethylated (33.3%) in comparison with the control group.

### 3.6. Effects of Different Treatments on the Expression of VNN1, ACC, FAS, SCD, and SREBPQ Genes in Different Organs of Geese

As shown in Figure 4A, the expression of the *VNN1* gene in the AZA group was significantly increased in liver and heart tissues, compared with the control group (*p* < 0.05). The expression of the *VNN1* gene in the liver, kidney, heart, and spleen tissues in the betaine group was significantly lower than that of the AZA group (*p* < 0.05). The expression of VNN1 in the AZA+betaine group was significantly lower than that in the AZA group, in terms of the liver and heart tissues (*p* < 0.05). No significant difference in *VNN1* gene expression in lung tissues was seen between the control group and other groups (*p* > 0.05). Figure 4B presents no significant effect on the expression of *FAS* gene (*p* > 0.05). Figure 4C shows no significant effect on the expression of the *ACC* gene in the liver, kidney, heart, and lung tissues (*p* > 0.05). The expression of the ACC gene in the spleen in the AZA group and the AZA+betaine group was significantly higher than that in the control group (*p* < 0.05). The feed of betaine significantly reduced the expression of the *ACC* gene (*p* < 0.05). According to Figure 4D, for the expression of the *SCD* gene in the liver, kidney, lung, and spleen, the betaine group showed significantly lower levels than the AZA group (*p* < 0.05). The expression of the *SCD* gene in the kidney and spleen was significantly higher in AZA+betaine group than in the betaine group (*p* < 0.05). Figure 4E shows that in terms of the liver and kidney, the expression of *SREBP* gene in the betaine group was significantly lower than that in the AZA group (*p* < 0.05). As to the expression of the *SREBP* gene in the kidney, the AZA group and AZA+betaine group showed a significantly higher level than the control group and betaine group (*p* < 0.05). There was no significant difference in the expression of the *SREBP* gene in the liver, heart, lung, and spleen compared with the control group (*p* > 0.05).

## 4. Discussion

As a nutritional supplement, betaine has demonstrated the capability to enhance weight gain in poultry. Chen et al. [21] reported that the body weights of the groups treated with 500 and 1000 mg/kg of betaine were significantly higher than those of the control group. A previous study indicated that betaine increased the body weight of 42-day-old geese [14]. However, the study also showed that the direct addition of betaine did not affect the body weight of geese. A prior research study showed that the addition of 0.35 mg/kg AZA, a methylation inhibitor, decreased body weight in mice [22]. Consistent with these findings, our study revealed that treatment with AZA significantly reduced the body weight of geese, while the co-administration of betaine and AZA did not significantly alter body weight. Therefore, a new assumption is suggested that betaine may exert its effects only under specific circumstances.

The levels of TG and TC in serum are recognized as important indicators of fat metabolism. HDL is responsible for the transport of TC from peripheral tissues and blood to the liver for further metabolism, HDL facilitates the transport of TC from peripheral tissues and blood to the liver for metabolism, while LDL is responsible for delivering TC synthesized in liver to extrahepatic tissues [23]. Most TG in the liver is transported to adipose tissue in the form of very low-density lipoprotein via the bloodstream. Our previous studies have found that the addition of betaine to the diet significantly reduced the levels of TG and TC in the serum of geese aged 63 days [14]. Additionally, a related research study showed that incorporating methyl donors into the diet can markedly decrease TG levels in primary chicken hepatocytes [24]. Leng et al. [25] demonstrated that betaine supplementation significantly lowered the levels of TG, TC, and HDL in serum. However, some studies have reported that treating cells with AZA alone did not affect TG levels [26]. In our study, the serum levels of TG and TC increased significantly in geese treated with AZA. This difference may be attributed to the change in experimental subjects and AZA treatment methodologies. In this experiment, the addition of betaine significantly reduced the content of TC in serum, with the content of TG reducing in a downward but not significant trend. These findings align with numerous previous studies that have clearly shown that betaine can enhance the rate of TG decomposition, inhibit TC synthesis, and reduce fat deposition in the body.

Betaine serves as an effective methyl donor. With SAM as the methyl donor, under the action of DNMT, SAH is generated after providing one carbon unit, and SAH is converted to homocysteine (Hcy) [27]. The ratio of SAM to SAH plays a critical role in regulating DNA methylation, as DNMT activity is contingent upon the levels of SAM produced and SAH consumed. When the ratio of SAM to SAH increases, systemic DNA hypermethylation occurs [28]. Accordingly, the SAM/SAH ratio is a key indicator for assessing the extent of DNA methylation [29]. A research indicated that betaine supplementation reduced SAH levels and increased the SAM/SAH ratio [30]. This study also demonstrated that the addition of betaine significantly elevated the SAM/SAH ratio. A similar investigation showed that dietary betaine supplementation in mice can up-regulate the SAM content and the hepatic SAM/SAH ratio in mice [31]. Betaine is a crucial methyl donor in the body. Through the catalytic reaction of BHMT (betaine-homocysteine-methyltransferase), methyl is transferred to homocysteine, resulting in the production of methionine, which subsequently generates SAM [32]. SAM serves as the primary methyl donor for DNA methyltransferases (DNMTs), suggesting that betaine might influence the methylation status of the *VNN1* gene promoter region by enhancing BHMT expression and increasing SAM production. In the rat model, betaine significantly affected the methylation level of spermatogenesis-related genes by up-regulating the activity of methyltransferase [33]. Conversely, AZA binds to DNA and inhibits DNMT activity to achieve the purpose of DNA demethylation [34]. The reduction in the SAM/SAH ratio and DNMT activity following AZA treatment can reverse the hypermethylation induced by betaine, decrease the level of DNA methylation, and promote fat deposition.

In general, the degree of DNA methylation is negatively correlated with the level of gene expression. Our previous study demonstrated that betaine can enhance DNA methylation in the promoter region of the VNN1 gene in hepatocytes and significantly reduced its expression [35]. 5-AZA is a classical DNA methyltransferase inhibitor, which inhibits its methyl transfer activity by forming a covalent complex with DNMTs, thereby achieving DNA demethylation [36]. In this study, 5-AZA was used to induce the high expression of the *VNN1* gene, and the DNA methylation level in the promoter region of *VNN1* was decreased. This finding further corroborates that betaine, as a methyl donor, can elevate DNA methylation levels in the promoter region of *VNN1*, thereby negatively regulating the gene expression of the latter. However, the use of 5-AZA may also bring some potential adverse effects. High concentrations of 5-AZA can lead to cytotoxicity and affect the normal physiological function of cells [37]. It has been demonstrated that the proliferation of endometrial stromal cells was significantly reduced and decidualization was impaired in mice [38]. The authors speculated that the cytotoxicity of 5-AZA was due to its ability to cause DNA damage.

To further elucidate the relationship between gene expression and DNA methylation, we also investigated the effects of AZA on the expression of fatty acid synthesis-related genes, including *FAS*, *ACC*, *SCD*, and *SREBPQ1*. The first three genes express the most important enzymes in fatty acid synthesis, while SREBP primarily regulates fatty acid synthesis at the transcriptional level [39]. Compared to the control group, the expression levels of *FAS*, *ACC*, *SCD*, and *SREBPQ1* genes in the betaine group were reduced to varying extents. Similarly, in broilers, the incorporation of betaine into the feed significantly decreased *FAS* gene expression in the liver [40]. A related study indicated that the addition of methyl donors to the diet of mice can lead to hypermethylation of the *FAS* gene promoter and consequently inhibition of the gene expression [41]. Furthermore, another research study showed that adding methyl donors diminished the translation and transcription of ACC and thereby inhibited the gene expression [42]. A similar trend was observed in *SREBP-1c* expression in mice [43]. In this study, treatment with AZA increased the gene expression of *FAS*, *ACC*, *SCD*, and *SREBPQ1*, further indicating that the elevated expression of these genes is associated with DNA demethylation. A previous study found that AZA treatment enhanced the expression of 5-fluorouracil by demethylation [44]. The findings from these experiments confirm that DNA methylation is negatively correlated with gene expression, a trend consistent with the results obtained in this study.

In summary, betaine has been widely used in poultry nutrition. In this study, betaine affected the expression of fat synthesis-related genes by regulating the DNA methylation of *VNN1* in goose liver fat metabolism, thereby reducing fat deposition. This finding provides a new idea for improving poultry meat quality through nutritional intervention.

## 5. Conclusions

The results of this study demonstrated that the increased expression of genes involved in fatty acid synthesis was related to the AZA-induced demethylation of the *VNN1* gene. This finding verified the mediating role of *VNN1* and its methylation in the mechanism of betaine inhibiting goose liver fat synthesis and provided new insights into the molecular basis of fat metabolism regulation.

## Figures and Tables

**Figure 1 animals-15-00719-f001:**
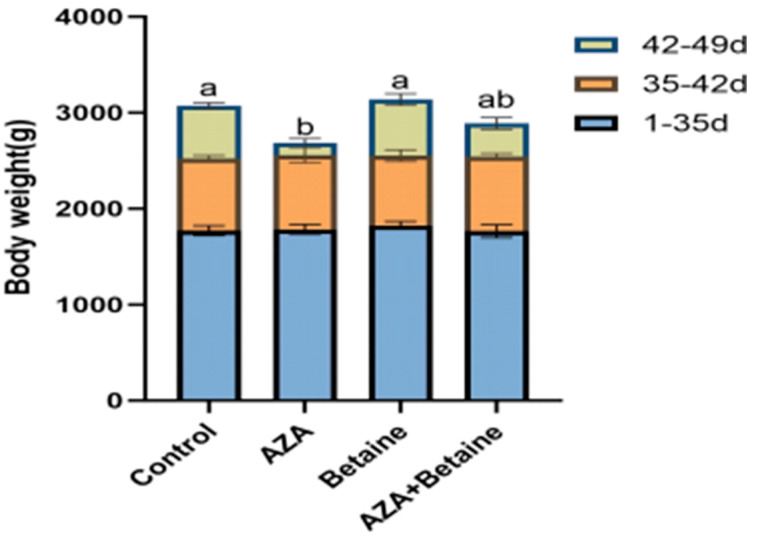
Effects of different supplemental treatments on the body weight of geese from 35 to 49 days of age. (1) The control group was treated with normal saline intraperitoneally (I.P.); the AZA group was treated I.P. with AZA (2 mg/kg); the betaine group was fed with betaine through the diet and treated I.P. with normal saline (1.2 g/kg); the AZA+betaine group was fed with betaine through the diet and treated I.P. with AZA. (2) Values are presented as means and standard error of the means (*n* = 7). Values followed by different letters superscript indicate significant differences (*p* < 0.05).

**Figure 2 animals-15-00719-f002:**
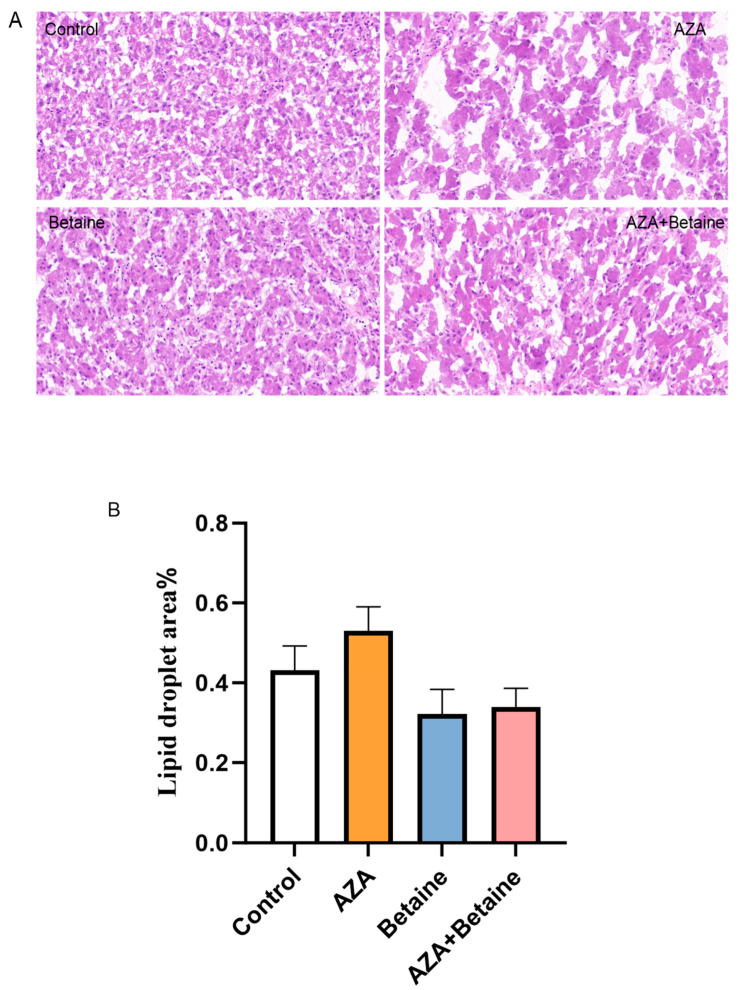
(**A**): Effects of different treatment groups on lipid morphology of geese liver. (**B**): The proportion of lipid droplet area in different treatment groups. (1) The control group was treated with normal saline intraperitoneally (I.P.); the AZA group was treated I.P. with AZA (2 mg/kg); the betaine group was fed with betaine through the diet and treated I.P. with normal saline (1.2 g/kg); the AZA+betaine group was fed with betaine through the diet and treated I.P. with AZA; (2) HE staining of liver lipid (40×); (3) each group contains 3 individuals. The mean± SEM is presented for each group (*n* = 3).

**Figure 3 animals-15-00719-f003:**
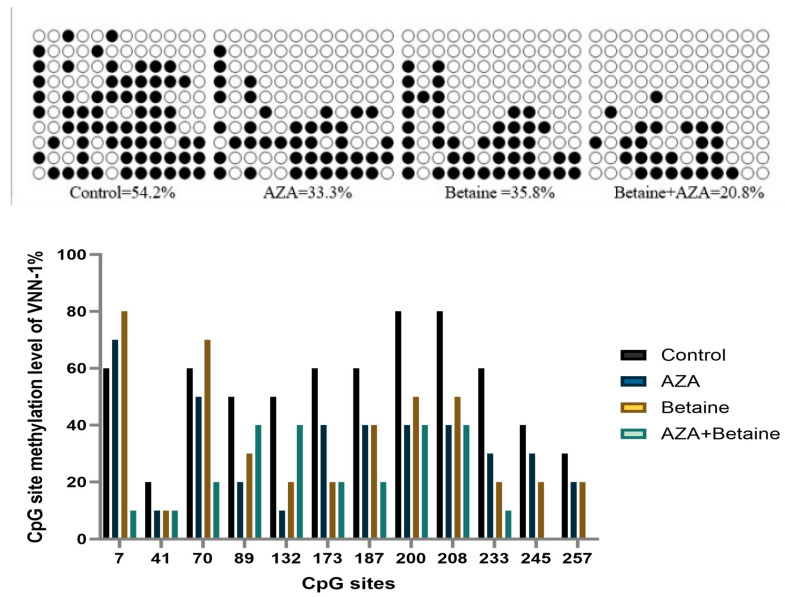
Effects of different treatments on DNA methylation level in VNN1 promoter region of gosling liver. (1) The control group was treated with normal saline intraperitoneally (I.P.); the AZA group was treated I.P. with AZA (2 mg/kg); the betaine group was fed with betaine through the diet and treated I.P. with normal saline (1.2 g/kg); the AZA+betaine group was fed with betaine through the diet and treated I.P. with AZA. (2) Closed and open circles represent methylated and unmethylated cytosines, respectively. Each bar represents the methylation level of the CpG site. Methylation at each CpG site of the promoter region was calculated by analyzing 10 clones.

**Figure 4 animals-15-00719-f004:**
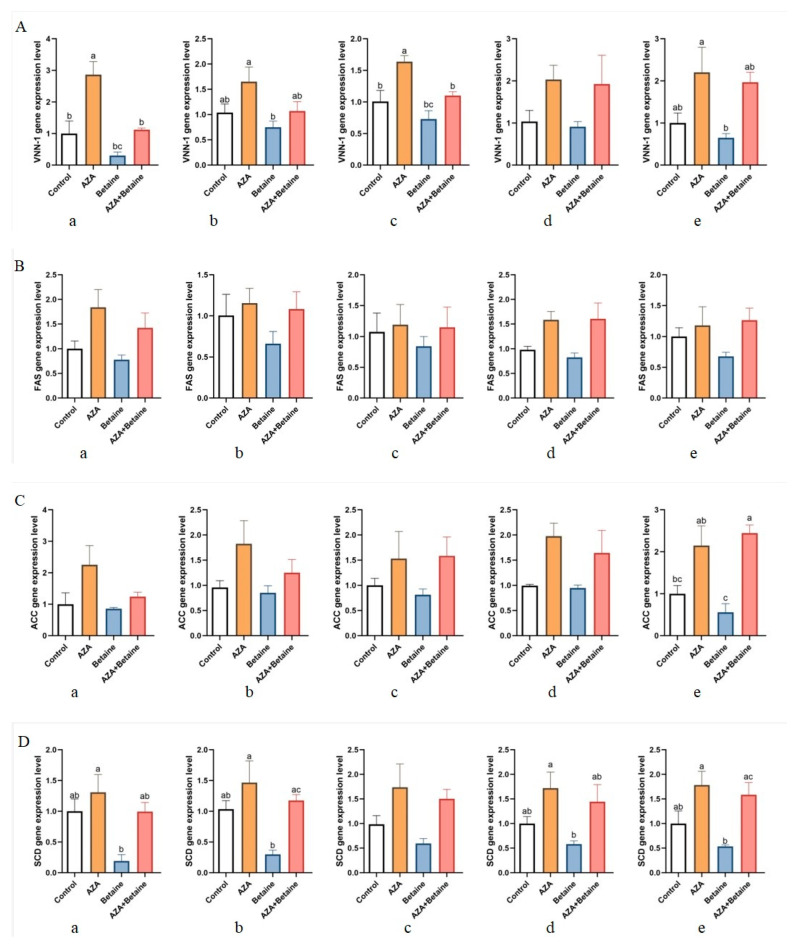
Effects of different treatment groups on the expression of *VNN1*, *FAS*, *ACC*, *SCD*, and *SREBPQ* genes in various organs. (1) (**A**) Effects of different treatments on the expression of VNN1 gene in the liver; (**B**) effects of different treatments on the expression of FAS gene in the kidney; (**C**) effects of different treatments on the expression of ACC gene in the heart; (**D**) effects of different treatments on the expression of SCD gene in the lung; (**E**) effects of different treatments on the expression of SREBPQ gene in the spleen; (2) values are presented as means and standard error of the means (*n* = 7). Values followed by different letters superscript indicate significant differences (*p* < 0.05).

**Table 1 animals-15-00719-t001:** Composition and nutrient levels of the experimental diets (air-dried basis %).

Ingredients	%
Corn	62.31
Soybean meal	29.27
Wheat bran	2.57
Rice husk	2.16
Limestone	0.90
Calcium hydrogen phosphate	1.27
DL-methionine	0.19
Lysine	0.03
Salt	0.30
Vitamin and trace mineral premix ^1^	1.00
Total	100.00
Nutritional level ^2^	
Metabolizable energy (MJ/kg)	11.55
Crude protein (%)	18.11
Crude fiber (%)	4.28
Lysine (%)	0.97
Methionine (%)	0.48
Calcium (%)	0.82
Available phosphorus (%)	0.38

^1^ One kilogram of premix contained: Vitamin A (transretinal acetate) 1,200,000 IU, Vitamin D3 (cholecalciferol) 400,000 IU, Vitamin E (tocopherol) 1800 IU, Vitamin K (antihemorrhagic) 150 mg, Vitamin B1 (thiamine) 60 mg, Vitamin B2 (riboflavin) 600 mg, Vitamin B6 (pyridoxamine) 200 mg, Vitamin B12 (cobalamin) 1 mg, nicotinic acid 3000 mg, pantothenic acid 900 mg, folic acid 50 mg, choline 35 g, biotin 4 mg, Fe 6 g, Cu 1 g, Mn 9.5 g, Zn 9 g, I 50 mg, Se 30 mg. ^2^ Metabolic energy, calcium and available phosphorus were calculated values, the rest were measured values.

**Table 2 animals-15-00719-t002:** Primer sequences for real-time PCR.

Gene	Primer Sequence (5′-3′)	Product Size (bp)	Reference
*β* *-actin*	F: GAAATCGTGCGTGACATCAA R: GCAGGACTCCAT*ACC*CAAGA F:	198	XM_066977989.1
*VNN1*	GAACTCAAGCACGAAGATGGA A R: CAGGTCTGTGCTCCTACACTTG A F:	189	XM-048075766.1
*FAS*	ATGCTTCAGGAGATGGGTATTG R: CCATCAGTGTTACTCCCAGCA F: TCCAGCAGA*ACC*GCATTGACA	118	XM-048050305.1
*ACC*	C R:GTATGAGCAGGCAGGACTTG GC	187	XM-048074039.1
*SCD*	F: TAACGGCTGGATCTCATCGC R: AGAGAACTTGTGGTGGACGC	149	XM-048069234.1
*SREBPQ*	F: GGTCCGGGCCATGTTGA R: CAGGTTGGTGCGGGTGA	175	XM-048066212.1

**Table 3 animals-15-00719-t003:** Effects of different treatments on the serum of geese ^1^.

Groups ^2^	Betaine (g/kg)	AZA (mg/kg)	TC (mmol/L)	TG (mmol/L)	HDL (mmol/L)	LDL (mmol/L)
A	0	0	4.15 ^b^	0.48 ^bc^	1.80 ^a^	1.42 ^b^
B	0	2	4.50 ^a^	0.70 ^a^	1.29 ^b^	1.83 ^a^
C	1.2	0	3.50 ^c^	0.44 ^c^	2.00 ^a^	1.29 ^b^
D	1.2	2	4.30 ^bc^	0.61 ^ab^	1.32 ^b^	1.59 ^b^
Betaine	SEM		0.10	0.03	0.09	0.06
	0		4.32 ^a^	0.59	1.55	1.66 ^a^
	1.2		3.90 ^b^	0.52	1.66	1.44 ^b^
AZA		0	3.82 ^b^	0.46 ^b^	1.90 ^a^	1.36 ^b^
		2	4.40 ^a^	0.66 ^b^	1.30 ^b^	1.71 ^b^
*p*-value		Betaine	<0.001	0.058	0.313	0.023
		AZA	<0.001	<0.001	<0.001	<0.001
		Betaine × AZA	0.008	0.385	0.478	0.431

^a–c^ Means with different superscripts within the same column indicate statistically significant difference (*p* < 0.05). ^1^ The data are presented as the mean ± SEM with *n* = 7 per treatment. TG, triglyceride; TC, total cholesterol; HDL, high-density lipoprotein cholesterol; LDL, low-density lipoprotein. ^2^ Group A was treated with normal saline intraperitoneally (I.P.); Group B was treated I.P. with AZA (2 mg/kg); Group C was fed with betaine through the diet and treated I.P. with normal saline (1.2 g/kg); Group D was fed with betaine through the diet and treated I.P. with AZA.

**Table 4 animals-15-00719-t004:** Effects of different treatments on enzyme activity in liver and serum of geese ^1^.

Groups ^2^	Betaine (g/kg)	AZA (mg/kg)	VNN1 (pg/mL)	SAM/SAH	DNMT (U/L)
A	0	0	17.93 ^b^	4.64 ^b^	109.24 ^ab^
B	0	2	21.09 ^a^	3.93 ^c^	80.86 ^c^
C	1.2	0	17.04 ^c^	5.06 ^a^	115.77 ^a^
D	1.2	2	19.54 ^ab^	4.56 ^b^	105.84 ^b^
Betaine	SEM		0.47	0.11	3.51
	0		19.51	0.86 ^b^	95.05 ^b^
	1.2		18.29	0.96 ^a^	110.81 ^a^
AZA		0	17.48 ^b^	0.97 ^a^	112.50 ^a^
		2	20.32 ^a^	0.85 ^b^	93.35 ^b^
*p*-value		Betaine	0.05	<0.001	<0.001
		AZA	<0.001	<0.001	<0.001
		Betaine × AZA	0.562	0.260	<0.001

^a–c^ Means with different superscripts within the same column indicate statistically significant difference (*p* < 0.05). ^1^ The data are presented as the mean ± SEM with *n* = 7 per treatment. VNN1, vanin-1; SAM/SAH, S-adenosylmethionine/S-adenosylhomocysteine; DNMT, DNA methyltransferase. The activity of VNN1 enzyme in the liver, SAM/SAH, and DNMT in serum were measured by ELISA kit. ^2^ Group A was treated with normal saline intraperitoneally (I.P.); Group B was treated I.P. with AZA (2 mg/kg); Group C was fed with betaine through the diet and treated I.P. with normal saline (1.2 g/kg); Group D was fed with betaine through the diet and treated I.P. with AZA.

## Data Availability

This review is based on data published in peer-reviewed journals many of which are open access.

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
