# Peer review of "Role of Vanin-1 Gene Methylation in Fat Synthesis in Goose Liver: Effects of Betaine and 5-Azacytidine Treatments"

_animals, 2025, doi:10.3390/ani15050719_

Round 1

Reviewer 1 Report

Comments and Suggestions for Authors

The author established four different treatment experimental groups to explore the impact of betaine and AZA on the weight and serum indexes of geese. Next, the activities of DNA methylation related enzymes in the goose liver were detected. Finally, the expression levels of genes related to fat metabolism in the geese were investigated. Although the study involved substantial work with detailed content, but the research methods require further improvement and some data are not convinced.

1. For the abstract, the logical of these sentence and please rewrite it.

2. Line 121, please confirm the force of centrifugation.

3. Line 139 and Line 145, absence of the number of subtitles.

4. In the figure 1, the gain weight of goose was over 5 kg for two weeks. Is it right? In general, the average gain weight of goose was about 100~200 g per day.

5. Line 223, the number of subtitle.

6. For the result “Effects of different treatments on fat morphology of goose liver”, the images quality of liver section is poor and please replace these images. Also, the fat morphology is difficult to observe and the oil red o staining is a better way to detect lipid accumulation. Also, the TG content analysis of liver was advised to add to address this issue in this work.

7. In the figure 3, whether the methylation level of VNN gene has the biological repeats, and there is absence of error bar in this chart.

8. In the figure 3, why did the authors detect the lipid metabolism genes in kidney, lung, spleen? They cannot support the liver lipid metabolism. In addition, if the authors detect more tissues, the adipose tissue might be a good choice.

Reviewer 2 Report

Comments and Suggestions for Authors

In this article, the authors determine that the vanin-1 (VNN1) gene methylation is associated with betaine-regulated fat synthesis in goose liver. Previous studies have implicated the importance of VNN1 gene in lipid metabolism. This study fills some of the existing gaps in the regulation and function of VNN1. The authors tested how Betaine treatment, along with a DNA methylate inhibitor alter the DNA-methylation of VNN1, and how that regulates fatty acid synthesis genes in goose liver. The research is done well, except the following comment.

Minor comments:

1.     In figure 1, the graph is a bit confusing. It would be better to use a scatter plot or plot the weight change.

Reviewer 3 Report

Comments and Suggestions for Authors

Title 

The title is informative but overly specific. Consider rephrasing it to: "Role of Vanin-1 Gene Methylation in Fat Synthesis in Goose Liver: Effects of Betaine and 5-Azacytidine Treatments" 

Line 13-15: unnecessary sentences that are creating confusion ‘Our previous study identified VNN1 as a key candidate gene that may be involved in the betaine-induced down-regulation of betaine in goose-liver fat synthesis. However, the exact regulatory mechanism of VNN1 in the process remains unknown’ 

Line 19-21: only this sentence is valid in simple summary ‘It was found that VNN1 and its methylation play a role in 19 betaine regulation of goose liver fat synthesis.’ authors are advised to rewrite the whole summary  

Abstract 

Authors are advised to write materials and methods in this section for better understanding  

Many things are not understandable in abstract terms. Authors are advised to write full terms on first seen when appear in abstract. Similarly, authors made a lot of unnecessary abbreviations in abstract which are not used later in abstract section.  

The conclusion is general. Highlight specific applications or implications of the findings. 

Introduction  

Introduction is poorly written. It needs to be rewritten to explain the research gap. Furthermore, hypothesis could be more explicitly linked to its practical significance, e.g., "understanding VNN1 methylation could lead to nutritional strategies for controlling fatty liver in ...

Line 37-38: do not claim anything without references ‘Excessive abdominal fat deposition in broiler poultry including goose has become an increasingly prominent issue in the industry’ add relevant references  

Line 44: these are not recent studies ‘Recent studies in mice have highlighted...’ 

Line 47: other study or studies ‘Moreover, other studies...’ 

Line 49-53: rewrite the sentence ‘These findings underscore the importance of the VNN1 gene in liver lipid metabolism. However, current researches primarily emphasize the effects of the VNN1 gene on lipid metabolism, leaving its functions and regulatory mechanisms largely unexplored. Therefore, the role of the VNN1 gene in poultry, particularly in geese, requires further investigation’ 

Materials and Methods 

May I know the feed type that was used in the study? 

Could you explain that why you did not check feed intake that my influence the body weight and hence overall objective of your study. This is the main point where your all expriment results could be disturbed  

The methods for DNA methylation analysis lack detail. Specify the kit or reagents used, and include manufacturer details for reproducibility 

Results 

The methylation data is intriguing but needs clearer visualization in figures. Please, include a heatmap or bar graph showing methylation levels across groups. 

"Gene expression levels were upregulated in..."  Specify the fold-change values for clarity 

Discussion 

More detail discussion is required. Please discuss potential molecular pathways through which betaine influences VNN1 methylation. Please also include comparisons with similar studies in other species. 

In discussion also include the role of 5-AZA in demethylation and potential adverse effects should be acknowledged. 

Expand on the practical applications of these findings in poultry nutrition or disease prevention. 

Comments on the Quality of English Language

Poor 

Round 2

Reviewer 3 Report

Comments and Suggestions for Authors

Thanks for your reply and effort to improve the quality of the manuscript